# Chronic Prostatitis/Chronic Pain Pelvic Syndrome and Male Infertility

**DOI:** 10.3390/life13081700

**Published:** 2023-08-07

**Authors:** Andrea Graziani, Giuseppe Grande, Michel Martin, Giordana Ferraioli, Elena Colonnello, Massimo Iafrate, Fabrizio Dal Moro, Alberto Ferlin

**Affiliations:** 1Unit of Andrology and Reproductive Medicine, Department of Medicine, University of Padova, 35122 Padova, Italy; andrea.graziani.3@studenti.unipd.it (A.G.); giuseppe.grande@aopd.veneto.it (G.G.); michel.martin@studenti.unipd.it (M.M.); 2Urology Clinic, Department of Surgery, Oncology and Gastroenterology, University of Padova, 35122 Padova, Italy; giordana.ferraioli@gmail.com (G.F.); massimo.iafrate@unipd.it (M.I.); fabrizio.dalmoro@unipd.it (F.D.M.); 3Department of Experimental Medicine, Sapienza University of Rome, 00185 Rome, Italy; elena.colonnello@uniroma1.it; 4Chair of Endocrinology and Medical Sexology (ENDOSEX), University of Tor Vergata, 00133 Rome, Italy

**Keywords:** chronic prostatitis, chronic pelvic pain syndrome, male infertility

## Abstract

Chronic prostatitis/chronic pelvic pain syndrome (CP/CPPS) is defined as urologic pain or discomfort in the pelvic region, associated with urinary symptoms and/or sexual dysfunction, lasting for at least 3 of the previous 6 months. The rate of symptoms related to prostatitis has a mean prevalence of 8–8.2%. CP/CPPS is most frequent in men younger than 50 years, among whom it is the most common urologic diagnosis. In the last decades, many studies have been published on CP/CPPS and its association with male infertility. The pathophysiologic relation between CP/CPPS and male infertility involves several aspects, which are not well studied yet. A reduction in semen parameters has been demonstrated in patients with CP/CPPS, and several mechanisms have been proposed to represent putative pathophysiological links between CP/CPPS and infertility, including male accessory gland inflammation, metabolic syndrome, inflammatory bowel disease, HPV co-infection and autoimmunity. In light of this evidence, a multidisciplinary approach is advocated for patients with known CP/CPPS, and particular attention is needed for male patients of infertile couples in order to evaluate male accessory glands correctly. In addition, it is advisable that future studies dealing with the treatment of CP/CPPS take into consideration all the different pathophysiological aspects implicated.

## 1. Introduction

Lower urinary tract symptoms (LUTS), pelvic pain and sexual symptoms caused by prostatic diseases severely affect the quality of life (QoL) of a considerable number of men. In fact, the prevalence of prostatitis and prostatitis-like symptoms is very high, and it might be comparable with the prevalence of ischemic heart disease and diabetes mellitus [1]. The definition of prostatitis can be traced back to Stamey, who also stated that it is a frequent and frustrating problem for all physicians [2]. The rate of symptoms which might be related to prostatitis ranges from 2.2 to 9.7% of men, with a mean prevalence of 8–8.2% [1,3,4]. Another study reported a 35–50% prevalence of prostatitis in men during their lifetime [5]. Even though prostatitis has been largely studied in the last decades, there is no generally accepted definition of such a condition or defined criteria for the diagnosis, nor is there a proper therapy [3]. Regarding the prevalence of prostatic disease in young men, the pathogenic role of prostatic calculi (PCal) has to be named. In fact, PCal have one of the highest prevalence rates in males, being present in about 50% of men and being correlated with the prevalence of prostatic disease and chronic prostatitis [6].

According to the National Institute of Health (NIH) consensus, prostatitis syndromes include infectious forms (acute, ABP, and chronic, CBP), chronic prostatitis/chronic pelvic pain syndrome (CP/CPPS) and asymptomatic prostatitis (AIP) [7].

Acute bacterial prostatitis (ABP), which represents <1% of cases of prostatitis, is caused by bacteria ascending from the urine to the urethra up to the prostate, with urine culture being considered the only laboratory evaluation of the lower urinary tract that is required [8,9]. Almost 10% of patients with ABP, despite optimal medical treatment, develop one of the chronic forms of prostatitis [10]. While several risk factors for progression to chronic bacterial prostatitis (CBP) have been identified—such as alcohol abuse, diabetes mellitus, large prostate volume and history of prostate surgery [11]—most of the patients with ABP who progress to CBP do not present known risk factors of progression. Gram-positive bacteria and atypical bacteria are the main pathogens involved in the etiology of CBP, with Enterococcus faecalis (46.90%), Staphylococcus (22.30%), Escherichia coli (15.09%) and atypical bacteria (6.04%) being the more prevalent [12].

On the other hand, most men receiving a diagnosis of CP/CPPS have no evidence of a genitourinary bacterial infection, and the cause is usually unknown [9]. In fact, <10% of men with chronic prostatitis have a proven bacterial infection [1,8].

Nowadays, CP/CPPS represents a perplexing challenge in everyday clinical practice [13], being associated with chronic pain, reduced QoL and, not ultimately, male infertility. However, the exact mechanisms connecting CP/CPPS and male infertility are not well clarified, nor is the prevalence of infertility in men with such conditions and, in turn, the recurrence of CP/CPPS among males presenting with infertility. The aim of this review is to focus on the association between CP/CPPS and male infertility, expanding on, in particular, the mechanisms involved in both these conditions.

### 1.1. Chronic Prostatitis/Chronic Pelvic Pain Syndrome (CP/CPPS)

*Chronic prostatitis/chronic pelvic pain syndrome (CP/CPPS*) (NIH category III) is defined as the presence of urologic pain or discomfort in the pelvic region, associated with urinary symptoms and/or sexual dysfunction, lasting for at least 3 of the previous 6 months [1]. Differential diagnoses include urinary tract infections (UTIs), cancer, anatomic abnormalities, gastrointestinal disorders, muscular disorders or neurologic disorders [1]. CP/CPPS might be subclassified into inflammatory type (NIH category IIIA) and noninflammatory type (NIH category IIIB), according to the presence of leukocytes in prostatic samples. Therefore, the presence of leukocytes confirms CP/CPPS type IIIA, whereas in type IIIB, there are no observed signs of inflammation [1,14]. In addition to clinical signs and symptoms of CP/CPPS, transrectal ultrasound (TRUS) might be helpful. In particular, ultrasound features of CP/CPPS include, among others, the presence of moderate or severe nonhomogeneity, hypoechoic texture, the presence of hyperemia and the presence of higher arterial prostatic peak systolic velocity [15].

CP/CPPS can occur in men of any age, but it is most frequent in men younger than 50 years, among whom it is the most common urologic diagnosis [16]. An Italian multicenter case-control observational study [17] reported a high prevalence of CP/CPPS (746 patients out of 5540 male urological outpatients, 13.5%) and found a close relation between CP/CPPS and cigarette smoking, high caloric diet, gastrointestinal or anorectal disease and multiple sexual partners.

Presenting symptoms in patients with IIIA- and IIIB-prostatitis are indistinguishable. The most common symptom is perineal, suprapubic, testicular and penile pain. Other symptoms are recurrent/intermittent irritative voiding symptoms (such as frequency, urgency and nocturia), obstructive symptoms (such as hesitation, dribbling, slow stream and retention) and dysuria. Indeed, prostatitis symptoms may overlap with those of benign prostatic hyperplasia (BPH) [18]. Painful ejaculation is also frequent, reported to affect more than half of men with CP/CPPS [19,20]. Sexual symptoms such as erectile dysfunction and ejaculatory dysfunctions such as premature ejaculation frequently occur [1,3,21,22].

As previously stated, CP/CPPS can negatively impact QoL and lead to negative behavioral consequences. Moreover, men with a previous history of sexual, physical or emotional abuse are more likely to have symptoms suggesting CP/CPPS. Therefore, some authors suggest screening patients with CP/CPPS for psychosocial symptoms [5]. In addition, the exacerbations, or “flares”, of CP/CPPS symptoms can worsen the outcome of other illnesses and might be, in future, considered an additional outcome measure in CP/CPPS research [23].

Systemic symptoms such as a low-grade fever may be present. Furthermore, the burden of pain in patients with CP/CPPS has also recently been underlined by the identification of functional and structural brain alterations in those patients [24]. Indeed, a lot of inflammatory mediator molecules and pain channels are involved in prostatitis-derived pain [25], often requiring the management of a pain specialist.

Since CP/CPPS is considered a multifactorial disease with many causes, a symptom-related approach is needed. For this reason, the NIH-Chronic Prostatitis Symptom Index (NIH-CPSI) and the UPOINT/UPOINTS concepts were developed [3]. The NIH-CPSI is a nine-item index developed to assess symptoms and QoL in men with CP/CPPS. The NIH-CPSI score embraces three subscores reflecting pain, urinary symptoms and QoL [26]. The UPOINT system can be employed to identify clinical phenotypes of patients. Moreover, it can be used to personalize therapy. In fact, the UPOINT system divides CP/CPPS symptoms into six categories: urinary (U); psychosocial (P); organ-specific (O); infection (I); neurologic/systemic (N); and tenderness (T). Each category has its suggested treatment [27]. The UPOINT system was validated to correlate with symptoms’ burden, and therapies guided by UPOINT were shown to lead to significant improvement of symptoms in 75–84% of patients [27]. Recently, an implementation of the UPOINT system to the UPOINTS system has been proposed. A domain related to sexuality (S), in fact, was added. As previously stated, sexual dysfunction is very frequent in CP/CPPS, with a high impact on the QoL of these patients [3]. Thus, the UPOINT classification represents the first holistic and all-inclusive approach to patients affected by CP/CPPS, permitting evaluation of the entire clinical picture of this complex disease. In addition, the International Prostate Symptom Score (IPSS) and the International Index of Erectile Function (IIEF) represent valuable measures to evaluate the condition and its response to treatment [1].

The mainstay of treatment for CBP continues to be antibiotic therapy. Combinations of antibiotics include fluoroquinolones for 4–6 weeks, trimethoprim-sulfamethoxazole, doxycycline and plurifloxacin plus serenoa repens plus arbutin plus lactobacillus sporogenes [4]. A longer treatment duration, between 6 and 12 weeks, is often necessary to achieve pathogen eradication [28]. In patients with persistent or recurrent symptoms and positive urine–sperm cultures, a low-dose daily prophylactic antibiotic can prevent symptoms from flaring up [29]. Long-term antibiotic therapy requires carefulness regarding the potential toxicity of each specific antibiotic, drug–drug interaction and risk of ECG-QT-interval elongation [30]. On the contrary, the treatment of CP/CPPS is not as well-defined and effective as the treatment for CBP. Monotherapy often is not satisfying, and tailored therapy, as suggested by the use of the UPOINTS system, is more efficacious [31]. Therapies employed in the treatment of CP/CPPS include physiotherapy and lifestyle modification. Moreover, patients with CP/CPPS can be treated with low-dose daily antibiotics, alpha-1 antagonists, 5-alpha reductase inhibitors, anti-inflammatory agents, phytotherapy and neuromodulatory agents. N-acetylcysteine could be useful as an adjunct to alpha-blockers, too [32]. Nevertheless, a recent review highlighted how pharmacological treatments had little evidence supporting their efficacy in CP/CPPS [33]. An interesting role in improving symptoms and QoL in patients with CP/CPPS is represented by acupuncture [1,4,22,27]. Moreover, recently, it was shown how low-intensity extracorporeal shockwave therapy (ESWT) is a safe, noninvasive and effective option for patients with CP/CPPS [34]. A recent clinical study investigating the combined role of ESWT, bromelain and escin in 95 patients with CP/CPPS reported an improvement in IPSS and NIH-CPSI results after 12 weeks of this combination therapy [35], despite having certain limitations, such as the lack of a placebo.

### 1.2. Male Factor Infertility

Couple infertility is defined as the inability to conceive after at least 12 months of regular unprotected sexual intercourse [36]. Infertility affects 15–20% of couples worldwide, and a male factor, per se, is responsible for about 30% of infertility cases and an additional 20% as a contributing cause [37,38]. The causes and risk factors for male infertility are numerous and can determine pretesticular (hypothalamic or pituitary impairment, with low gonadotropin levels), testicular (characterized by high gonadotropin levels) and post-testicular (with gonadotropin levels generally in the normal range) forms [36,39]. Importantly, other than the classic major causes (such as, for instance, cryptorchidism, chemotherapy, testicular cancer or genetic factors), risk factors for male infertility are related to lifestyle and different medical conditions, including cardiovascular, metabolic and other chronic diseases [36,38,40]. Indeed, male factor infertility, and, in general, the fertility potential of a man are considered a mirror of the general health of a man [41].

History, physical examination and semen analysis represent the first-line and most relevant diagnostic tests to gather information on the fertility potential of a man, and their results guide the following diagnostic process. Semen analysis must be performed following the *WHO Manual*, 6th edition, recently renewed [42], and should report the classical parameters (such as semen volume, pH, sperm concentration, total sperm count, sperm motility and morphology), which should be interpreted together to give clinical significance. Other second-line tests on semen are suggested in selected cases. In particular, semen reactive oxidative species (ROS) and the sperm DNA fragmentation index (DFI) [36] could give information on specific forms of male infertility. In fact, an excessive amount of ROS is related to male infertility [43], since an increase in oxidative stress in semen is linked to a decrease in sperm quality and vitality [44]. It is also known that chronic diseases, such as obesity and diabetes, increase the production of endogenous ROS [43], creating a vicious link between these chronic diseases and male infertility. Regarding DFI, increasing evidence indicates that it represents a marker of damaged chromatin and sperm function, with a role in male infertility and reproductive success independent from standard sperm parameters [45]. Interestingly, although the etiologies of increased sperm DNA fragmentation are numerous, oxidative stress is recognized as a major factor.

In cases of suspected male genital tract infection or inflammation contributing to male infertility, sperm culture and urethral swab or urine culture for Chlamydia trachomatis and Mycoplasmas are recommended [39]. Finally, first-line exams include testicular ultrasound in all male partners of infertile couples and TRUS prostate–vesicular scan in patients with suspected distal duct obstruction or abnormalities and in patients with genital tract infection and inflammation [39]. Indeed, ultrasound exams represent fundamental diagnostic tests in the management of male factor infertility. A recent study [46] showed a higher prevalence of prostate and seminal vesicle alterations in infertile men with normal FSH plasma concentrations (<8 U/L) with respect to infertile men with high FSH concentrations (≥8 U/L). Furthermore, ultrasound abnormalities correlate with seminal parameters and give important information on genital tract characteristics.

## 2. Relation between CP/CPPS and Male Factor Infertility

In the last decades, many studies have been published on CP/CPPS [47,48] and its association with male infertility. The evidence that infertile men have higher prostate-specific antigen (PSA) concentrations than fertile individuals represents a crucial aspect suggesting a possible prostatic involvement in the etiology of male infertility [49]. Indeed, the pathophysiologic relationship between CP/CPPS and male infertility involves several aspects.

A systematic review and meta-analysis conducted in 2017 by Condorelli et al. including 27 studies with a total of 3241 participants found that CP/CPPS was associated with standard parameters of semen analysis, such as reduction in semen volume, sperm concentration, progressive motility and normal morphology [50]. Furthermore, CP/CPPS has been associated with an increased risk of antisperm antibodies (ASAs) in semen [50] and functional parameters of sperm, such as higher levels of DNA fragmentation and changes in protamine mRNA ratio compared to healthy individuals [51].

Such associations with semen and sperm parameters in patients with CP/CPPS might have different pathophysiological mechanisms. Therefore, we conducted a literature review on this topic and selected the 11 studies available (among 607 results found using the search terms “prostatitis and infertility”) published in the last ten years (January 2012–December 2022) in English, representing clinical studies, systematic reviews and meta-analyses and excluding non-English results, case reports, reviews, opinion papers and results without available manuscripts. These studies are reported in Table 1. A brief analysis and discussion will follow, with the aim of exploring the different pathophysiological mechanisms linking CP/CPPS and male infertility.

A 2012 transversal study by Hou DS et al. [52], evaluating 51 men with CP/CPPS, 11 infertile men and 26 controls, proposed that expressed prostatic secretions (EPS)-related bacteria are a critical actor in the development of CP/CPPS and infertility, with most of the EPS from CP/CPPS and infertility patients found to be 16S rRNA gene-positive. In our literature review, we found no other significant studies regarding EPS, CP/CPPS and infertility. A meta-analysis conducted in 2014 by Shang Y et al. [55] that considered seven studies involving 249 patients with CBP and 153 controls reported a significant negative effect of CBP on sperm vitality, sperm total motility and the percentage of progressively motile sperm. The meta-analysis seems to confirm the association between male infertility and CBP, although its mechanisms are not completely elucidated. Among the proposed mechanisms, accessory gland inflammation is a key element linking CP/CPPS and infertility. A transversal study [15] evaluated 400 men with prostatitis-like symptoms and reported that patients with prostatitis-like symptoms have higher semen IL-8 levels and that a higher prevalence of prostate ultrasound alterations is observed in subjects with a higher NIH-CPSI total score. A transversal study [54], enrolling 169 infertile patients (with CBP and/or bilateral prostato-vesiculitis (BPV)) and 42 controls, showed that seminal hyperviscosity is associated with increased oxidative stress in infertile men and increased pro-inflammatory interleukins in patients with male accessory gland infection/inflammations, more when the infection is extended to the seminal vesicles. Schagdarsurengin U et al. [59], enrolling 105 NIH IIIb-CP/CPPS patients and 41 controls, found that CP/CPPS has adverse effects on surrogate male fertility parameters, such as sperm motility, morphology and semen pH. In addition, a systematic review and meta-analysis [50] found similar results, with CP/CPPS related to the reduction in semen volume, concentration, progressive motility, normal morphology and zinc concentration in seminal plasma. Moreover, CP seemed to significantly increase the risk of developing ASAs on seminal plasma, as reported in the 2016 meta-analysis [57]. Furthermore, a transversal study [51] studying 41 patients with CP/CPPS and 22 controls reported results in contrast to the previously obtained ones; in particular CP/CPPS patients were found to have decreased ejaculate volumes, pH, motility, normal morphology and vitality, increased levels of IL-8 and increased DNA fragmentation index (DFI) levels compared to the control group. Another transversal study conducted by Lotti F et al. [56] reported an increase in the number of metabolic syndrome (MetS) components negatively associated with normal sperm morphology and positively with serum IL-8 levels, prostate total and transitional zone volume, arterial peak systolic velocity, texture nonhomogeneity and calcification size. In 2012, Vicari et al. [53] analyzed 50 patients with CBP and IBS, 56 infertile men and 30 controls and reported reduced total sperm count and motility and higher seminal leukocyte concentration in patients with CBP plus irritable bowel syndrome (IBS) or CBP alone than in the fertile men. While not dealing with infertility in CP/CPPS, the study appears to be noteworthy in light of the evidence of a relationship between chronic prostatitis and IBS. It has been proposed that HPV co-infection could represent another important link between CBP and infertility. Cai T et al. found that co-infection of *Chlamydia trachomatis* and HPV decreases male fertility, in particular regarding sperm motility and morphology [54]. Those data might indicate a role of HPV in the genesis of CP/CPPS-induced infertility. Finally, a 2016 systematic review and meta-analysis [57] identifying six studies with 721 patients with CP/CPPS and 160 controls found a significant correlation of the antisperm antibodies (ASAs)-positive relationship between patients with CP/CPPS and controls.

In conclusion, a reduction in semen parameters has been demonstrated in patients with CP/CPPS, and several mechanisms have been proposed to represent putative pathophysiological links between CP/CPPS and infertility, including male accessory gland inflammation, metabolic syndrome, IBS, HPV co-infection and autoimmunity.

### 2.1. Male Accessory Gland Inflammation (MAGI)

CP/CPPS can be considered as the top of the iceberg of male accessory gland infection/inflammation (MAGI). According to the World Health Organization (WHO) criteria, MAGI is diagnosed in patients with oligo-, astheno- and/or teratozoospermia associated with at least one factor A (history of genitourinary infection or physical signs) plus one factor B (abnormality of prostatic fluid), one factor A plus one factor C (ejaculate signs, such as leukocyte >1 million/mL, culture with significant growth of pathogenic bacteria, abnormal appearance, increased viscosity, increased pH and/or abnormal biochemistry of the seminal plasma), one factor B plus one factor C or two factors C [60].

While the definition developed by the WHO considers the presence of sperm abnormalities to be a key prerequisite for the diagnosis of MAGI, the NIH classification of prostatitis does not consider sperm analysis a crucial step in the diagnostic process [61]. This discrepancy might have led to an underestimation of the prevalence of semen alterations in patients with prostatitis, especially patients with CP/CPPS. The prevalence of MAGI is reported from 5 to 30%, but frequently it is underestimated because it is often asymptomatic or paucisymptomatic [61]. According to the number of involved glands, MAGI might be divided into uncomplicated (prostatitis) or complicated (prostato-vesiculitis and prostato-vesiculo-epididymitis), and, depending on its extension, into unilateral or bilateral forms [61].

The frequency of MAGI among infertile patients ranges from 2 to 18% according to the diagnostic criteria used [61]. Leukocytospermia, polymorphonuclear elastase and the analysis of the secretory products of male accessory glands have been widely used in the past for diagnosis of MAGI, although their diagnostic significance is uncertain [60]. Furthermore, some cytokines, such as IL-6 and IL-8, might be promising markers for the diagnosis and follow-up of MAGI [62].

More recently, seminal protein markers of MAGI have been proposed [60]. In detail, seminal soluble urokinase-type plasminogen activator receptor (suPAR) has been demonstrated to be a marker of bacterial and abacterial MAGI [63]. Moreover, an increase in seminal suPAR has been reported in abacterial MAGI associated with male hypogonadism [64] and psoriasis [65].

Importantly, the TRUS prostate–vesicular scan is essential in infertile males with suspected MAGI to assess its extension and the prostate and seminal vesicles echo-pattern [39]. It is also important to note the promising role of magnetic resonance (MRI) as a diagnostic tool in patients with CP/CPPS [66], although its use in infertile males is not recommended.

MAGI can compromise male fertility through the production of ROS and/or inflammatory cytokines, impaired secretory capacity of the accessory glands, anatomical obstruction or subobstruction of the seminal tract and direct effect of microorganisms on spermatozoa [61].

The secretory dysfunction of male accessory glands, triggered by microorganisms or by the inflammatory response, might be the most important cause of MAGI-related negative impact on sperm conventional and biofunctional parameters. Higher numbers of glands involved are associated with the worst sperm parameters and a more severe symptomatology. Although all patients with MAGI have a higher risk for infertility, both bilateral prostato-vesiculo-epididymitis and the fibro-sclerotic variant represent the MAGI categories with the worst sperm parameters [61].

ROS can be produced either by leukocytes in seminal fluid, which increase during inflammation or infection, or by the spermatozoa themselves. This production, through the phenomenon of lipid peroxidation, results in damage to sperm and mitochondrial membranes, resulting in altered motility and sperm nuclear and mitochondrial DNA [67]. The correlation between abacterial MAGI and fertility has been demonstrated by the efficacy of anti-inflammatory treatment in improving seminal parameters in these patients [68].

### 2.2. Metabolic Syndrome

Metabolic syndrome is a cluster of risk factors for cardiovascular and metabolic diseases, including increased triglycerides, increased blood pressure, central adiposity, increased fasting glucose and decreased high-density lipoprotein cholesterol [69]. Obesity could represent a central element for both male infertility and CP. It is well-known that obesity and its underlying mediators have a negative impact on semen parameters, including sperm concentration, motility, viability and normal morphology [70]. Obesity and metabolic syndrome, in fact, have been associated with male infertility, hypogonadism and sexual dysfunctions [71,72].

On the other hand, metabolic syndrome, obesity, insulin resistance and diabetes mellitus [73] could be associated with CP and represent risk factors for CP, as well as LUTS, development [74]. Indeed, obesity and metabolic syndrome represent a chronic inflammatory disease, mediated by a complex underlying pathophysiology where inflammatory cytokines result in subclinical prostatitis and prostatic hyperplasia [70].

### 2.3. Impaired Microcirculation and Vascularization

Several studies reported an association between varicocele and CP [75] and between varicocele, CP and premature ejaculation [76]. Starting from these premises, it can be speculated that varicocele, leading to intrapelvic congestion and dilation of the periprostatic venous plexus, might impair prostate microcirculation, thus causing prostatic inflammation. Gat et al. [77] demonstrated the presence of a venous blood reflux from the high-pressure testicular venous drainage system to the low-pressure prostatic drainage system through a direct communication represented by the deferential vein and the vesicular plexus. The presence of communication between the testicular and the prostatic venous system might justify a back-flow of venous blood from the testis to the prostate, which can lead to intrapelvic venous congestion. This could facilitate the onset of symptoms of prostatitis. Accordingly, it has been demonstrated that the selective occlusion of impaired venous drainage in the male reproductive system is associated with a reduction in prostate volume and benign prostatic hyperplasia-related symptoms [77].

### 2.4. Autoimmunity and Inflammation

Inflammation certainly has a main role in the pathogenesis of CP. It also creates an important link between CP and male infertility, further underlined by the possible autoimmune etiology of CP [78,79]. Moreover, prostate inflammation has an important major role in determining sperm quality [80].

The possible role of autoimmunity on the fertility status in patients with CP/CPPS is exemplified by the finding of antisperm antibodies (ASAs) [81]. ASAs might be directed against different portions of the sperm and might interfere with sperm motility and transport through the female reproductive tract, as well as with capacitation and acrosome reactions [50]. A systematic review and meta-analysis, conducted by Jiang et al. in 2016 [57], reported six studies on the association between ASAs and CP/CPPS and concluded that a higher ASA level was present in CP patients than in healthy people, although well-designed studies are necessary to further study this issue. The mechanism leading to ASA formation seems to be related to alteration of the blood–testis barrier, which isolates spermatogenesis from the immune system. When the integrity of the genitourinary tract is damaged, such as by inflammation induced in CP, the immune system might be stimulated and ASAs formed. In addition, semen alterations in patients with CP/CPPS have been related to autoimmune response against different prostate antigens [82].

Independent of autoantibodies formation, inflammation and related cytokines, as well as reactive oxygen species (ROS), might be important for fertility in men with CP/CPP. For example, inflammatory mediators and ROS seem to be involved in the observed acrosomal dysfunction associated with CP [83]. Th17 and Th1 cells and related cytokines IL-17 and IFN-γ seem to be the main actors in the autoimmune injury of CP/CPPS. Secretions of IFN-γ and IL-17 were reported to be higher in patients with CP/CPPS than controls, and patients with CP/CPPS had significantly greater serum immunoglobulin G immune reactivity to seminal plasma proteins [84]. The same authors reported how inflammation of the genital tract was proved by the presence of high levels of IFNγ, IL-17, IL-1β,IL-8 and macrophages in the semen and that this local inflammation was associated with an overall diminished semen quality and higher levels of sperm apoptosis/necrosis in patients with CP/CPPS [84]. Moreover, recent data indicate that Treg cells in CP/CPPS patients may have functional defects [85]. Importantly, this inflammatory state could contribute to the explanation of the significant increase in prostate cancer risk in patients with CP [30,86,87]. It has also been shown that hyaluronan (HA), an extracellular matrix component which accumulates in chronic inflammatory tissues, is elevated in patients with CP/CPPS and that targeting HA through 4-methylumbelliferone decreases the proportion of Th1 cells relieving CP [88]. Finally, another study, using experimental autoimmune prostatitis models, indicated CXCL10 as an important mediator involved in inflammatory infiltration and pain symptoms of prostatitis by promoting the migration of macrophages and secretion of inflammatory mediators [89]. Finally, recent evidence showed that CP was significantly associated with reduced zinc concentration in seminal plasma, and that seems to be crucial for semen alteration [50]. Therefore, nowadays there is an important focus on the role of the immune system and inflammation in CP/CPPS and MI, in particular considering the perspective of biological immunology and immune microenvironment [85]. A recent study on rats reported that pirfenidone—an anti-inflammatory, antifibrotic and antioxidative stress drug—significantly ameliorated chronic pelvic pain and inhibited prostatic inflammation and fibrosis by suppressing the expression of pro-inflammatory mediators; exhibiting a mighty antioxidant capacity by improving the activities of glutathione, catalase and total superoxide dismutase; and inducing the polarization of M2 macrophages and suppression of the activation of the nuclear factor-κB (NF-κB) signaling pathway [90]. To summarize the evidence, patients with CP/CPPS present Treg and Th17 cell dysfunction, abnormal regulation of T helper 1 and T helper 2 cells and macrophages, with their related cytokines playing a noteworthy role in the tissutal etiology of CP/CPPS. This evidence has been recently reported in a systematic review and meta-analysis, where, after the analysis of 34 studies, authors reported that TNF-α, IL-1β, IL-6 and IL-8 are the four immune mediators that were elevated in most of the samples derived from patients with CP/CPPS and the experimental autoimmune prostatitis models.

### 2.5. Irritable Bowel Syndrome (IBS)

Irritable bowel syndrome (IBS) is defined by the presence of abdominal pain associated with the alteration of bowel habits without an underlying structural pathology. IBS is a common condition that affects 9–23% of the general population with a considerable impact on quality of life and health care costs [91]. Several studies have been published in the last years on IBS, and the new Rome IV symptom-based criteria were published in 2016, bringing considerable changes in the diagnostic criteria for IBS [91].

The main symptoms of IBS include chronic abdominal pain and/or discomfort, diarrhea, constipation and bloating. These symptoms are present in some patients daily, while others have these symptoms intermittently, at intervals of weeks or, rarely, even months [92].

Interestingly, prostatitis syndromes and IBS are frequently associated [93]. A study enrolling 152 patients with prostatitis syndrome and 204 patients with IBS, conducted by Vicari et al. [93], reported a prevalence of 31.2% of both prostatitis syndrome and IBS. Those patients had total NIH-CPSI and pain subscale scores higher than patients with prostatitis alone and a higher prevalence of CBP.

Mechanisms underlying both these conditions include primitive pathogenetic factors (such as intestinal dysbiosis) or secondary pathogenetic mechanisms due to modified commensal gut flora related to antibiotic therapy, which in turn may lead to mucosal inflammation [93]. In detail, gut dysbiosis has been correlated with worse seminal parameters [94]. Furthermore, patients with prostatitis have seminal dysbiosis, characterized by a reduction in Lactobacilli and an increase in Protobacteria [78]. Although these studies need further confirmation, it is worth noting that male infertility is more prevalent in patients with IBS than in the general population [75].

### 2.6. Human Papillomavirus

Infection by human papillomavirus (HPV) has been advocated as another etiology of infertility in patients with CP [95]. La Vignera et al. proved that ultrasound signs of prostatitis more frequently occurred in patients with evidence of HPV-DNA presence in the semen, especially in those with high-risk genotypes [96]. Another study showed that in patients with prostatitis-related symptoms attributable to Chlamydia trachomatis, infection and co-infection with HPV had a significant role in decreasing male fertility, in particular with regard to sperm motility and morphology [54]. Therefore, as proposed, it is possible that HPV infection may be associated with the degree of intraprostatic inflammation [97]. It is important to underline that the prevalence of HPV semen infection ranges between 2 and 31% in men from the general population and between 10 and 35.7% in men affected by infertility, mainly asthenozoospermia [98]. Moreover, HPV infection has been associated with the development of ASAs [99]. Indeed, this fascinating topic related to the possible association between CP/CPPS, male infertility and HPV infection needs further studies.

## 3. Final Considerations

The literature is rich in possible associations between CP/CPPS and male infertility, although a certain connection between those conditions has not been found yet. Different mechanisms have been proposed in order to explain this association. In this manuscript, we present the most studied mechanisms of the possible association between CP/CPPS and male infertility. The association between CP/CPPS and male infertility also represents one of the five “hot spots” regarding prostatitis research trends over the last decades. In light of this premise, and in light of the evidence presented above, some important aspects of future research are represented by the study of the gut microbiome and the study of the inflammatory state, considering inflammasome, autoimmunity, inflammation itself, chemokines and cytokines. Those might be considered biomarkers of both CP/CPPS and male infertility and possible targets for future research and treatment. In addition, if on the one hand, research regarding CP/CPPS is addressed towards the study of sperm DNA fragmentation and sperm protamine mRNA ratio, then, on the other hand, even the study of male infertility is addressed towards new approaches, which consider the study of genetic and postgenetic alterations in a multi-omics perspective and the study of oxidative stress and sperm DNA fragmentation [100,101]. Finally, last but not less important is the awareness of this disease. In fact, if CP/CPPS might be underestimated in its prevalence and its reduction of QoL, male infertility is well-associated with lack of awareness and reduced self-motivation to seek medical help, even in recent studies [102].

## 4. Conclusions

CP/CPPS is a very common disease that causes severe symptoms and can negatively impact QoL. Cumulative evidence indicates that CP/CPPS has a detrimental effect on sexual function, sperm parameters and male infertility. The relationship between prostatitis and male factor infertility is linked to several etiological and pathophysiological associations such as MAGI, autoimmunity, HPV co-infection, obesity and metabolic syndrome and IBS. Therefore, a multidisciplinary approach is advocated for patients with known CP/CPPS looking for pregnancy and particular attention is needed for male patients of infertile couples in order to correctly evaluate male accessory glands. This review did not consider therapeutic aspects of CP/CPPS in men with semen alteration and/or infertility, but it is advisable that future studies dealing with treatment take into consideration the different pathophysiological aspects.

## Figures and Tables

**Table 1 life-13-01700-t001:** Results from our literature PubMed review. Abbreviations: ASAs: antisperm antibodies; BPV: bilateral prostato-vesiculitis; CBP: chronic bacterial prostatitis; CP/CPPS: chronic prostatitis/chronic pain pelvic syndrome; DFI: DNA fragmentation index; EPS: expressed prostatic secretions; IBS: irritable bowel syndrome; MAGI: male accessory gland infection/inflammation; NIH-CPSI: NIH-Chronic Prostatitis Symptom Index.

Authors	Year	Type of Study	Patients	Results	Conclusion of the Authors
Hou, D.S. et al. [52]	2012	Transversal	- 51 CP/CPPS - 11 infertile - 26 controls	EPS bacterial community structure differs among the three groups of individuals, and the 16S rRNA gene-positive rate of the CP/CPPS patients is higher than that among the normal men. Most of the EPS from CP/CPPS and infertility patients are 16S rRNA gene-positive.	EPS-related bacteria play an important role in the development of CP/CPPS and infertility.
Vicari, E. et al. [53]	2012	Transversal	- 50 patients with CBP and IBS - 56 infertile - 30 controls	Total sperm count and motility are reduced and seminal leukocyte concentration is higher in patients with CBP plus IBS or CBP alone than in the fertile men. Seminal leukocyte concentrations are higher in the patients with CBP plus IBS compared with the patients with CBP alone.	IBS might be considered a comorbidity in CBP patients with a worse reproductive prognosis.
Cai, T. et al. [54]	2014	Transversal	- 1003 patients with chronic prostatitis-related symptoms	Sperm motility and normal morphology are lower in patients with HPV and *Chlamydia trachomatis* infection (group B) than in patients with *Chlamydia trachomatis* alone (group A). Moreover, 298 (41.6%) men in group A and 192 (66.8%) men in group B were subfertile.	In a population of prostatitis-like symptoms attributable to *Chlamydia trachomatis* infection, co-infection with HPV decreases male fertility, in particular regarding sperm motility and morphology.
Lotti, F. et al. [54]	2014	Transversal	- 400 men with prostatitis-like symptoms	Patients with prostatitis-like symptoms have higher semen IL-8 levels, while no differences in semen parameters are observed when comparing subjects with or without prostatitis-like symptoms. Furthermore, a higher prevalence of prostate moderate–severe nonhomogeneity, hypoechoic texture and hyperemia as well as a higher arterial prostatic peak systolic velocity is observed in subjects with a higher NIH-CPSI total score.	NIH-CPSI scores and prostatitis-like symptoms, evaluated in infertile men, are related not to sperm parameters but mainly to clinical and CDU signs of infection/inflammation.
Shang, Y. et al. [55]	2014	Meta-analysis	- Seven studies were considered, including 249 CBP patients and 153 controls	Sperm vitality, total motility and the percentage of progressively motile sperm from CBP patients were significantly lower than controls, although CBP had no significant effect on semen volume, sperm concentration and the duration of semen liquefaction.	There was a significant negative effect of CBP on sperm vitality, sperm total motility and the percentage of progressively motile sperm. Additional studies with a larger number of subjects are needed.
Lotti, F. et al. [56]	2014	Transversal	- 171 infertile, of whom only 44.4% (*n* = 76) were without components of MetS	An increase in the number of MetS components correlates negatively with normal sperm morphology and positively with serum IL-8 levels, prostate total and transitional zone volume, arterial peak systolic velocity, texture nonhomogeneity and calcification size.	In infertile males, MetS might be a trigger for a subclinical, early-onset form of benign prostatic hyperplasia.
Jiang, Y. et al. [57]	2016	Systematic review and meta-analysis	- Six studies were identified, including 721 CP/CPPS patients and 160 controls	There is a significant correlation of the ASA-positive relationship between CP patients and controls. The combined odds ratio of the ASA-positive rate in CP patients and normal controls was 3.26.	The positive rate of ASAs in CP patients is significantly higher than in the control group, suggesting that CP has a negative effect on male reproductive function.
Castiglione, R. et al. [58]	2014	Transversal	- 169 infertile patients (with CBP and/or BPV) - 42 controls	In BPV infertile patients, ROS generation and pro-inflammatory cytokines levels are higher than those found in CBP infertile patients and controls, although seminal IL-10 levels in BPV and CBP patients are lower than those found in the controls.	Seminal hyperviscosity is associated with increased oxidative stress in infertile men and increased pro-inflammatory interleukins in patients with MAGI, more when the infection is extended to the seminal vesicles.
Schagdarsurengin et al. [59]	2017	Transversal	- 105 NIH IIIb- CP/CPPS patients - 41 controls	CP/CPPS NIH IIIb is associated with significant alteration of sperm motility, morphology and semen pH. Patients older than 33 show increased seminal IL-8 levels.	NIH IIIb CP/CPPS has negative effects on surrogate parameters of male fertility.
Condorelli, R.A. et al. [50]	2017	Systematic reviews and meta-analysis	- 27 studies included, with 3241 participants (381 with CBP, 1670 with CP/CPPS and 1190 controls)	CBP is associated with reduction in sperm concentration, vitality and total and progressive motility, whereas CP/CPPS is related to the reduction in semen volume, concentration, progressive motility normal morphology and zinc concentration in seminal plasma. Moreover, CP statistically increases the risk of developing ASAs on seminal plasma.	CP has a detrimental effect on sperm, and both CPB or CP/CPPS might differently show negative impact on sperm.
Berg, E. et al. [51]	2021	Transversal	- 41 CP/CPPS - 22 controls	CP/CPPS patients have decreased ejaculate volumes, pH and total and progressive motility, normal morphology and vitality, increased amounts of peroxidase-positive cells and immature germ cells and higher levels of IL-8 compared to healthy controls. Moreover, CP/CPPS patients showed increased DFI levels compared to the control group.	There are significant differences concerning conventional semen parameters in CP/CPPS patients compared to healthy men. CP/CPPS patients have increased DFI, which emphasizes an unfavorable impact on fertility at a molecular level.

## Data Availability

Not applicable.

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
