# Peer review of "Chronic Prostatitis/Chronic Pain Pelvic Syndrome and Male Infertility"

_life, 2023, doi:10.3390/life13081700_

Round 1
Reviewer 1 Report
1. Please add the reference from Chinese (PMID: 31156012), which indicates that the 51.65% males suffered from chronic prostatitis/prostatic calculi.
2. The authors described more about the clinical findings, but lack of the potential mechanism in CP or male infertility, even in the part of inflammation. Recent days, the study of how immunocytes impact CP has been widely reported, for Th1, Th17, Treg and Macrophage, from patients to animal model. Based on the mechanism can provide more potential to reveal the relation between CP and male infertility, only clinical is not enough.
The language should be polished.
Author Response
Thank you for your suggestions. We added the suggested reference. Moreover, we added several information regarding the inflammation pathway and the mechanisms involved in the aetiology of CP/CPPS.
Reviewer 2 Report
The review is chaotic. It is difficult to read and understand the essence of what the authors want to say.
1. Introduction
Table 1 is not relevant to this review.
Chronic prostatitis is a disease that begins inside the prostate gland due to impaired microcirculation. There are no words about it. The recently developed concepts of UPOINT/UPOINTS mainly refer to chronic pelvic pain in men.
2. Relation between CP/CPPS and Male Factor Infertility
11 articles presented in table. 2 are not related to each other, and their subsequent explanation has no connection between them.
3. Conclusions
The conclusion is a continuation of the discussion.
References are disordered.
Author Response
We are grateful to the Reviewer for his/her observations. We erased Table 1. We added some informations regarding impaired prostatic microcirculation in a proper paragraph. Furthermore, we re-organized the presentation of the studies found in our literature review and proposed a new paragraph, called “final considerations”, in order to reduce the paragraph on conclusions.
Reviewer 3 Report
I believe it would be interesting to follow the correct methodology of a systematic review (obviously after recognised limitations of evidence in this field)
Extensively shown a review of literature but it should be interesting to show population, intervention, comparison and outcomes for each
Also GRADE system should used to evaluate quality of evidence
Correct English
Author Response
We are grateful to the Reviewer for his/her observations. Since GRADE system is used in clinical recommendations and/or guidelines, we were not able to use it in our review. We furthermore preferred to keep our manuscript as a narrative review, although improving the manuscript, adding some informations regarding the evidences described in each original study we evaluated.
Round 2
Reviewer 3 Report
Accept